# Intentions Regarding COVID-19 Vaccination in Females Aged 15–49 Years

**DOI:** 10.3390/vaccines10020336

**Published:** 2022-02-20

**Authors:** Shihoko Kajiwara, Naomi Akiyama, Michio Ohta

**Affiliations:** 1School of Nursing, Gifu University of Health Science, 2-92 Higashiuzura, Gifu 500-8281, Japan; n-akiyama@gifuhoken.ac.jp; 2School of Rehabilitation, Gifu University of Health Science, 2-92 Higashiuzura, Gifu 500-8281, Japan; m-oota@gifuhoken.ac.jp

**Keywords:** COVID-19, vaccine, intention, healthcare, promoting, hesitancy, Japan

## Abstract

To control the coronavirus disease 2019 (COVID-19) pandemic, the Japanese government is promoting vaccination, which many people are willing to accept; however, some are reluctant to receive vaccinations. The purpose of this study was to analyze the intentions of women aged 15–49 years regarding the COVID-19 vaccination and to identify methods of promoting vaccination. We used secondary data from a web research company of approximately 1020 participants. The data contained the following variables: vaccination status, reasons for not getting vaccinated, and the intentions and reasons related to the third vaccination. We categorized the reasons using text data and evaluated the age-related differences. The proportion of women aged 15–49 years who refused COVID-19 vaccination in Japan was 17.0%, and the rate was not significantly different by age group. The most common reasons were safety and side effect concerns. Of those who received the second vaccination, 32.7% hesitated or refused the third vaccination, and the rate was not significantly different by age group. The reasons were side-effect concerns, a lack of information, and the influence of their surroundings. Addressing the side effects and providing adequate information may help promote vaccination among women aged 15–49 years.

## 1. Introduction

The coronavirus disease 2019 (COVID-19) pandemic has changed people’s lives. In Japan, socially vulnerable women are said to experience increasing difficulties due to unemployment and reduced income. In fact, according to suicide statistics from the Ministry of Health, Labor and Welfare, the suicide rate among women and young people in 2020 has increased significantly [1]. To control the COVID-19 pandemic, vaccines are being developed rapidly and they are being promoted worldwide [2]. In Japan, vaccination of medical staff began in February 2021, and vaccination of people aged 12 years and over is currently underway. The Japanese government is also promoting the spread of rapid vaccination to restore people’s normal lives, and many people are willing to get vaccinated. However, a certain number of people are hesitant.

Many studies on vaccination intentions have been conducted because people’s willingness to be vaccinated is important to economic recovery [3,4,5,6,7,8,9,10,11,12,13,14,15,16,17]. According to these studies, females aged 20–49 years and low-income earners have low vaccine acceptance rates [3,4,5,6]. In other words, people who are vulnerable in society and most likely to desire socioeconomic recovery are reported to be less willing to get vaccinated. However, these surveys were conducted before COVID-19 vaccination started in Japan, and it has been said that vaccine hesitancy may change over time [7].

In Japan, approximately 10 months have passed since vaccinations started for people aged 12 years and over, and currently, a third booster vaccination of medical staff is underway. Therefore, this study analyzed the intentions of females aged 15–49 years regarding COVID-19 vaccinations. The purpose of this study was to derive ideas for promoting future vaccinations by analyzing intentions regarding COVID-19 vaccination. We found that the proportion of women aged 15–49 years who refused the COVID-19 vaccination in Japan was 17.0%; this rate was not significantly different by age group. Of those who completed the second vaccination, 32.7% hesitated or refused the third vaccination; this rate also did not significantly differ by age group.

## 2. Materials and Methods

In this study, anonymized data collected and provided by an internet research company were used. This study targeted females aged 15–49 years nationwide because young females are one of the characteristic populations with low vaccine acceptance rates, and addressing the concerns of this population group will help promote vaccination. The data of 1020 participants were evenly collected by age group (15–19 years, 20–29 years, 30–39 years, and 40–49 years).

Of the total, data of 1013 participants were analyzed, excluding 7 individuals due to inappropriate answers and dropping out from the study (Figure 1). The youngest age group was 15–19 years because only individuals aged 15 years and above are eligible to register as panel members of the internet research company. The survey was conducted in September 2021 in Japan.

### 2.1. Measures

COVID-19 vaccination status: COVID-19 vaccination status was assessed by the question “Did you get the COVID-19 vaccine?” The response options were: “My first vaccination is complete,” “My second vaccination is complete,” “Vaccination reservation not completed/I cannot make a reservation,” and “I will not get vaccinated in the first place.”

Reason for having no intention to get vaccinated: The respondents who answered “I will not get vaccinated in the first place” were asked to state their reason for providing this response (text answer).

Intention to accept the third vaccination: Those who answered, “My second vaccination is complete” were assessed by the question “If there is a next vaccination (third vaccination or next year’s annual vaccination, etc.), do you get vaccinated?” The response options were the following: “I will get vaccinated,” “I will not get vaccinated,” and “I am not sure.” The respondents were also asked to state their reason (text answer).

### 2.2. Analyses

Descriptive statistics were used to evaluate the following: (1) the proportion of those who answered, “I will not get vaccinated in the first place” and their reasons; (2) the intention to receive the third vaccination by those who answered “My second vaccination is complete”; (3) the reasons for giving the answer “I am not sure” or “I will not get vaccinated” regarding the third vaccination. Data were also analyzed using a chi-square test to compare differences by age group. All analyses were performed using JMP version 12.0 and *p* < 0.05 was considered statistically significant.

We categorized the text data related to the answer “I will not get vaccinated in the first place” into 11 categories: “I am allergic to vaccines/I have other physical reasons,” “I do not like needles/injections,” “I am not concerned about getting seriously ill from the coronavirus/I do not need it,” “I will not have time to get vaccinated,” “I would be concerned about getting infected with the coronavirus/other disease from the vaccine/I am worried about its safety,” “I am afraid/anxious,” “I would be concerned about side effects from the vaccine,” “I do not think vaccines work very well,” “The coronavirus outbreak is not as serious as some people say it is/There is no need for society,” “Blank/No reasons/Bad answer,” and “Others,” with reference to research conducted by Latkin et al. [8]. The text data related to the reasons for the responses “I will not get vaccinated” or “I am not sure” regarding the third vaccination was classified into 12 categories: the above-mentioned 11 categories and a 12th category, “It depends on what people around me do/There is not enough information to make a decision.” Then, the categories mentioned by the respondents as reasons were reclassified as “agree,” and the categories not mentioned as reasons as “disagree.” Categorization was performed and verified by two researchers.

This study was conducted in accordance with the ethical principles of the Declaration of Helsinki. The data used were secondary data, and the researchers did not have a correspondence table that was linked to the participants. The internet research company that provided the data conducts research using the panel platform of the panel owner company. Comprehensive consent was obtained from the panelists when they registered as members.

## 3. Results

### 3.1. Vaccination Status and Proportion of Those Who Answered “I Will Not Get Vaccinated in the First Place”

Of the 1013 participants analyzed, the following 172 participants (17.0%) answered that they would not get vaccinated in the first place: 44 participants (17.6%) aged 15–19 years, 46 (17.9%) aged 20–29 years, 46 (18.3%) aged 30–39 years, and 36 (14.2%) aged 40–49 years, without a significant difference by age group (*p* = 0.574; Table 1).

### 3.2. Reasons for Giving the Answer “I Will Not Get Vaccinated in the First Place”

The most common reasons given by the 172 respondents who answered, “I will not get vaccinated in the first place” were safety concerns categorized as, “I would be concerned about getting infected with the coronavirus/other disease from the vaccine/I am worried about its safety”, including “The future reactions from the vaccination are unknown/safety of the vaccine has not been confirmed” (*n* = 44, 25.6%), followed by side effects concerns categorized as, “I would be concerned about side effects from the vaccine”, including “I am afraid of side effects/I must take a break from work or school due to side effects” (*n* = 38, 22.1%). The third most common reasons were health problems categorized as, “I am allergic to vaccines/I have other physical reasons”, including “I am allergic/I am pregnant/I am sick” (*n* = 24, 14.0%).

The answers that showed significant differences between age groups were “I do not like needles/injections” (*p* < 0.001) and “I am afraid/anxious” (*p* = 0.002), and most of the answers were from the 15–19 year old group (Table 2).

### 3.3. Intention Regarding the Third Vaccination among Those Who Answered, “My Second Vaccination Is Complete”

Of the 350 respondents who answered that their second vaccination was completed, 348 responded to the question regarding the third vaccination. Of the 348 respondents, 234 (67.2%) answered “I will get vaccinated,” 12 (3.4%) answered “I will not get vaccinated,” and 102 (29.3%) answered “I am not sure.” There was no difference by age group (*p* = 0.212; Table 3).

### 3.4. Reasons for Giving the Answer, “I Will Not Get Vaccinated” or, “I Am Not Sure”, Regarding the Third Vaccination

Of the 350 respondents who answered that the second vaccination was completed, approximately one-third (*n* = 114, 32.8%) answered “I will not get vaccinated” or “I am not sure” about the third vaccination. The most common reasons for the 114 respondents were side effect concerns, categorized as “I would be concerned about side effects from the vaccine,” including “Side effects were hard,” “Side effects disturbed my work,” and “I’m afraid of side effects” (*n* = 37, 32.5%). Furthermore, 26 respondents (22.8%) said “It depends on what people around me do/There is not enough information to make a decision” and “It depends on infection status.” However, there was no difference by age group (*p* = 0.858, *p* = 0.277; Table 4).

## 4. Major Discussion

Our results presented three major conclusions.

First, we revealed that approximately 17% of women aged 15–49 years clearly refused vaccination, with no significant difference by age group. Excluding 257 participants who had not completed vaccination or could not make a reservation, the proportion of those who had been vaccinated once or twice and those who did not intend to be vaccinated was 77.3% and 22.7%, respectively. Compared to a survey conducted before vaccinations started in Japan, Yoda et al. reported that 12.3% of Japanese people, 11.8% aged 49 years and under, did not want to get vaccinated, and vaccination intentions differed by age group [3]. In addition, Kadoya et al. reported that 22.0% of Japanese people, including 26.7% of those aged 21–50 years old, were unwilling to be vaccinated and said that their unwillingness and hesitation would change over time [7]. In the United States, Latkin et al. reported that 16.7% of Americans and 24.3% of 18–39-year-olds will not receive the coronavirus vaccine, and that vaccination intentions differed by age group [8]. However, these studies did not target young women and instead included a wide range of age groups; it has been found that women are less willing to be vaccinated. In our findings, the proportion of participants who clearly refused vaccination was in the range of 11.8–26.7%, similar to that reported in the pre-vaccination study, although there was a slight difference in the age range used in the analysis. Regarding the difference in the intention to be vaccinated by age group, our results differed from those of the previous study. This is probably because we targeted a narrow age range of females (15–49 years).

Second, the most common reason for giving the answer “I will not get vaccinated in the first place” was safety concerns categorized as “I would be concerned about getting infected with the coronavirus/other disease from the vaccine/I am worried about its safety.” This result was similar to that of the study by Yoda et al., who reported that approximately two-thirds of those who hesitate or do not want to get vaccinated say, “I am concerned about potential side effects” and “I think the COVID-19 vaccine may not be safe” [3]. The difference in percentages is considered to be due to the difference in methods because we obtained the reasons as text data. In Europe, Neumann-Böhmeetal reported that 55% of those who hesitate to get vaccinated and 38% of those who do not want to get vaccinated are concerned about potential side effects, which is the most common reason [6]. In addition, in both previous studies, this concern was more common in females than in males. Young women may be concerned about the negative effects on pregnancy and childbirth in the next generation [18,19]. In Japan, the adverse events of the human papilloma virus (HPV) vaccine for cervical cancer have been extensively reported; it has been regarded as a major problem compared to other countries, and the vaccination rate has decreased significantly [3,20]. Therefore, many young women may be suspicious of the COVID-19 vaccine, which has not been used sufficiently, followed by side effect concerns categorized as “I would be concerned about side effects from the vaccine” (22.1%). The frequency and severity of side effects of the COVID-19 vaccine are higher than those of inactivated vaccines such as influenza [21], and it is said that side effects are more prominent in females than in males. As mentioned, for young women who are vulnerable in society, serious side effects are considered to be a great threat to their lives and cause them to refuse vaccination.

Third, of those who completed the second vaccination, 32.7% answered “I will not get vaccinated” or “I am not sure” regarding the third vaccination. The most common reason was side effect concerns (32.5%). Side effect concerns among those who answered “I will not get vaccinated in the first place” are, so to speak, imagined, but side effect concerns being the reason for hesitating to get a third vaccination are a result of experience, and this is considered a fear rather than actual concerns. The ratio of side effect concerns in the reason for hesitating to get a third vaccination was approximately 1.5 times that of those who answered, “I will not get vaccinated in the first place.” This suggests that actual adverse side effects due to COVID-19 vaccination may interfere with the promotion of vaccination and should be addressed. The second most common reason was “It depends on what people around me do/There is not enough information to make a decision” (22.8%). According to Yoda et al., a certain number, if not many, of people say, “I think that I should get vaccinated because people around me are doing so.” Conversely, the unwillingness to get vaccinated is considered to lead to an increasing tendency to refuse vaccination. It can be said that this is a Japanese-specific reason that is not often observed in other countries. This is because Japanese people tend to attach great importance to groupism rather than individualism. In addition, the lack of information to make a decision may be affected by current trends. Sufficient information must be provided to reduce vaccination refusal.

Sample size: the data of 1020 individuals were obtained. According to the demographics as of 1 October 2019, by the Statistics Bureau of the Ministry of Internal Affairs and Communications, the population of Japanese women aged 15–49 years was approximately 24 million. Assuming a margin of error of 5%, a confidence level of 95%, and a response distribution of 50% (i.e., maximum), the minimum sample size was calculated as 384 [22,23], and the amount of data obtained were sufficient.

Limitations: This study was web-based; hence, registered panelists may have biases regarding their place of residence, income, education level, etc. In addition, there may have been a tendency toward vaccination acceptance due to this being a characteristic of the panelists. For example, a person who is fluent on the Internet and is registered as a panelist at an Internet research company may have a higher income and educational level and may be able to collect a significant amount of information from the Internet. There are reports that income and educational level are related to the acceptance of the COVID-19 vaccine. Additionally, intentions regarding COVID-19 vaccination can be markedly affected by information. Therefore, these may affect the results of this study and may not completely align with the average Japanese vaccination tendency. However, since we used secondary data, detailed data on panelist attributes and social demographics were not evaluated. In addition, we will need to expand the research to a wider population, including men and older people who are more vulnerable to COVID-19 in a future study.

## 5. Conclusions

The proportion of women aged 15–49 years who have refused the COVID-19 vaccination in Japan was 17.0%, and the rate was not significantly different by age group. The most common reasons were safety and side effect concerns. Of those who completed the second vaccination, 32.7% hesitated or refused the third vaccination, and the rate was not significantly different by age group. The reasons for this were side effect concerns, a lack of information, and the influence of their surroundings.

Based on these findings, it is suggested that addressing side effects and providing sufficient information will lead to the promotion of vaccination acceptance among females aged 15–49 years.

## Figures and Tables

**Figure 1 vaccines-10-00336-f001:**
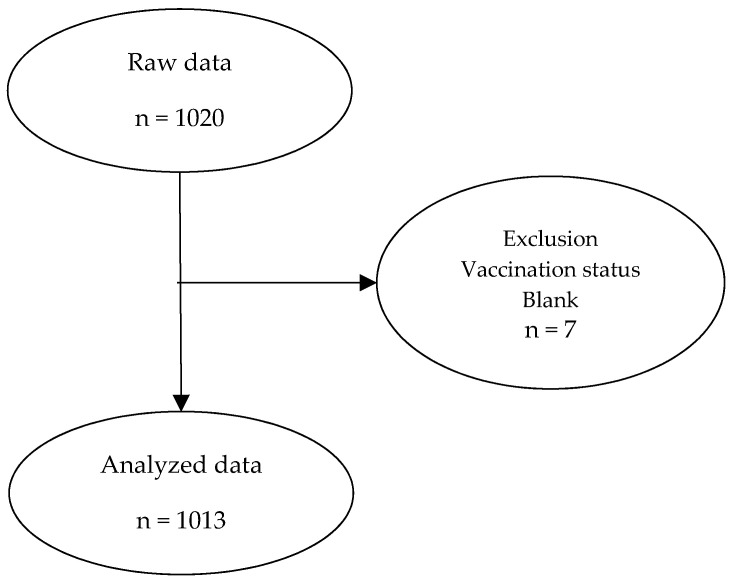
Analyzed data.

**Table 1 vaccines-10-00336-t001:** Vaccination status and proportion of those who answered, “I will not get vaccinated in the first place”.

	Total		15–19 Years		20–29 Years		30–39 Years		40–49 Years		*p*-Value
	*n* = 1013 (%)		*n* = 250 (%)		*n* = 257 (%)		*n* = 252 (%)		*n* = 254 (%)		
First vaccination is complete	234	(23.0)	59	(23.6)	53	(20.6)	61	(24.2)	61	(24.0)	0.003
Second vaccination is complete	350	(34.6)	65	(26.0)	87	(33.9)	88	(34.9)	110	(43.3)	
No reservation	257	(25.4)	82	(32.8)	71	(27.6)	57	(22.6)	47	(18.5)	
I will not get vaccinated	172	(17.0)	44	(17.6)	46	(17.9)	46	(18.3)	36	(14.2)	
I will not get vaccinated	172	(17.0)	44	(17.6)	46	(17.9)	46	(18.3)	36	(14.2)	0.574
Other	841	(83.0)	206	(82.4)	211	(82.1)	206	(81.7)	218	(85.8)	

*p*-value was calculated using the chi-square test.

**Table 2 vaccines-10-00336-t002:** Reasons for giving the answer, “I will not get vaccinated in the first place”.

	Total		15–19 Years		20–29 Years		30–39 Years		40–49 Years		*p*-Value
	*n* = 172 (%)		*n* = 44 (%)		*n* = 46 (%)		*n* = 46 (%)		*n* = 36 (%)		
I am allergic to vaccines/I have other physical reasons.											
Agree	24	(14.0)	5	(11.4)	5	(10.9)	6	(13.0)	8	(22.2)	0.481
Disagree	148	(86.0)	39	(88.6)	41	(89.1)	40	(87.0)	28	(77.8)	
I do not like needles/injections.											
Agree	12	(7.0)	9	(20.5)	2	(4.3)	0	(0.0)	1	(2.8)	<0.001
Disagree	160	(93.0)	35	(79.5)	44	(95.7)	46	(100.0)	35	(97.2)	
I am not concerned about getting seriously ill from the coronavirus/I do not need it.											
Agree	16	(9.3)	3	(6.8)	4	(8.7)	3	(6.5)	6	(16.7)	0.431
Disagree	156	(90.7)	41	(93.2)	42	(91.3)	43	(93.5)	30	(83.3)	
I will not have time to get vaccinated											
Agree	1	(0.6)	1	(2.3)	0	(0.0)	0	(0.0)	0	(0.0)	0.433
Disagree	171	(99.4)	43	(97.7)	46	(100.0)	46	(100.0)	36	(100.0)	
I would be concerned about getting infected with the coronavirus/other disease from the vaccine/I am worried about its safety.											
Agree	44	(25.6)	7	(15.9)	15	(32.6)	12	(26.1)	10	(27.8)	0.306
Disagree	128	(74.4)	37	(84.1)	31	(67.4)	34	(73.9)	26	(72.2)	
I am afraid/anxious.											
Agree	20	(11.6)	11	(25.0)	4	(8.7)	5	(10.9)	0	(0.0)	0.002
Disagree	152	(88.4)	33	(75.0)	42	(91.3)	41	(89.1)	36	(100.0)	
I would be concerned about side effects from the vaccine.											
Agree	38	(22.1)	13	(29.5)	9	(19.6)	11	(23.9)	5	(13.9)	0.359
Disagree	134	(77.9)	31	(70.5)	37	(80.4)	35	(76.1)	31	(86.1)	
I do not think vaccines work very well.											
Agree	11	(6.4)	1	(2.3)	4	(8.7)	4	(8.7)	2	(5.6)	0.490
Disagree	161	(93.6)	43	(97.7)	42	(91.3)	42	(91.3)	34	(94.4)	
The coronavirus outbreak is not as serious as some people say it is/There is no need for society.											
Agree	0	(0.0)	0	(0.0)	0	(0.0)	0	(0.0)	0	(0.0)	NA
Disagree	172	(100.0)	44	(100.0)	46	(100.0)	46	(100.0)	36	(100.0)	
Blank/No reasons/Bad answer											
Agree	20	(11.6)	3	(6.8)	6	(13.0)	7	(15.2)	4	(11.1)	0.621
Disagree	152	(88.4)	41	(93.2)	40	(87.0)	39	(84.8)	32	(88.9)	
Others											
Agree	9	(5.2)	4	(9.1)	1	(2.2)	3	(6.5)	1	(2.8)	0.413
Disagree	163	(94.8)	40	(90.9)	45	(97.8)	43	(93.5)	35	(97.2)	

*p*-value was calculated using the chi-square test. NA; not analyzed.

**Table 3 vaccines-10-00336-t003:** Intention regarding the third vaccination among those who answered, “My second vaccination is complete”.

	Total		15–19 Years		20–29 Years		30–39 Years		40–49 Years		*p*-Value
	*n* = 348 (%)		*n* = 65 (%)		*n* = 87 (%)		*n* = 88 (%)		*n* = 108 (%)		
I will get vaccinated.	234	(67.2)	38	(58.5)	55	(63.2)	65	(73.9)	76	(70.4)	0.212
I will not get vaccinated.	12	(3.5)	4	(6.1)	4	(4.6)	3	(3.4)	1	(0.9)	
I am not sure.	102	(29.3)	23	(35.4)	28	(32.2)	20	(22.7)	31	(28.7)	

*p*-value was calculated using the chi-square test.

**Table 4 vaccines-10-00336-t004:** Reasons for the response, “I will not get vaccinated” or, “I am not sure”, regarding the third vaccination.

	Total		15–19 Years		20–29 Years		30–39 Years		40–49 Years		*p*-Value
	*n* = 114 (%)		*n* = 27 (%)		*n* = 32 (%)		*n* = 23 (%)		*n* = 32 (%)		
I am allergic to vaccines/I have other physical reasons.											
Agree	1	(0.9)	0	(0.0)	0	(0.0)	0	(0.0)	1	(3.1)	0.464
Disagree	113	(99.1)	27	(100.0)	32	(100.0)	23	(100.0)	31	(96.9)	
I do not like needles/injections.											
Agree	1	(0.9)	0	0.0)	0	0.0)	1	(0.0)	0	(0.0)	0.357
Disagree	113	(99.1)	27	(100.0)	32	100.0)	22	(100.0)	32	(100.0)	
I am not concerned about getting seriously ill from the coronavirus/I do not need it.											
Agree	10	(8.8)	2	(7.4)	3	(9.4)	4	(17.4)	1	(3.1)	0.328
Disagree	104	(91.2)	25	(92.6)	29	(90.6)	19	(82.6)	31	(96.9)	
I will not have time to get vaccinated.											
Disagree	114	(100.0)	27	(100.0)	32	(100.0)	23	(100.0)	32	(100.0)	NA
I would be concerned about getting infected with the coronavirus/other disease from the vaccine/I am worried about its safety.											
Agree	8	(7.0)	3	(11.1)	1	(3.1)	2	(8.7)	2	(6.3)	0.650
Disagree	106	(93.0)	24	(88.9)	31	(96.9)	21	(91.3)	30	(93.8)	
I am afraid/anxious.											
Agree	1	(0.9)	0	(0.0)	0	(0.0)	1	(4.3)	0	(0.0)	0.357
Disagree	113	(99.1)	27	(100.0)	32	(100.0)	22	(95.7)	32	(100.0)	
I would be concerned about side effects from the vaccine.											
Agree	37	(32.5)	9	(33.3)	9	(28.1)	9	(39.1)	10	(31.2)	0.858
Disagree	77	(67.5)	18	(66.7)	23	(71.9)	14	(60.9)	22	(68.8)	
I do not think vaccines work very well.											
Agree	7	(6.1)	0	(0.0)	3	(9.4)	1	(4.3)	3	(9.4)	0.206
Disagree	107	(93.9)	27	(100.0)	29	(90.6)	22	(95.7)	29	(90.6)	
The coronavirus outbreak is not as serious as some people say it is/There is no need for society.											
Disagree	114	(100.0)	27	(100.0)	32	(100.0)	23	(100.0)	32	(100.0)	NA
Blank/No reasons/Bad answer											
Agree	16	(14.0)	2	(7.4)	9	(28.1)	1	(4.3)	4	(12.5)	0.049
Disagree	98	(86.0)	25	(92.6)	23	(71.9)	22	(95.7)	28	(87.5)	
Others											
Agree	15	(13.2)	3	(11.1)	3	(9.4)	1	(4.3)	8	(25.0)	0.121
Disagree	99	(86.8)	24	(88.9)	29	(90.6)	22	(95.7)	24	(75.0)	
It depends on what people around me do/There is not enough information to make a decision.											
	26	(22.8)	9	(33.3)	4	(12.5)	5	(21.7)	8	(25.0)	0.277
	88	(77.2)	18	(66.7)	28	(87.5)	18	(78.3)	24	(75.0)	

*p*-value was calculated using the chi-square test. NA; not analyzed.

## Data Availability

The data are available under reasonable request to the corresponding author.

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
