# Peer review of "Intentions Regarding COVID-19 Vaccination in Females Aged 15–49 Years"

_vaccines, 2022, doi:10.3390/vaccines10020336_

Round 1
Reviewer 1 Report
The reviewed ms aims to analyze the intentions of Japanese women aged 15–49 years to get vaccinated for COVID-19. The authors used secondary data from a web research company of > 1000 participants. The study reports a significant proportion of women aged 15–49 years who refused COVID-19 vaccination in Japan aprox. 17.0%, and the rate was not found significantly different by age group. The most common reasons were safety and side effect concerns. Of those who received the second vaccination, 32.7% hesitated or refused the third vaccination, and the rate was not significantly different by age group. Overall, the ms is good and the nicely presented data would be of interest to the reader of the MDPI Vaccines.
Specific comments and suggestions
I am not convinced that the presented data are sufficient to justify a full-length rather than a short report ms.
- It would be interesting to see the rates of men refused to vaccinate against COVID-19
- Not sure why authors focused only in Japanese women aged between 15 to 49 and not older ones, that consist a more vulnerable to COVID-19 group
- The comment above also stands for men
Author Response
February 4, 2022
Dear Mr./Ms. peer reviewer 1,
Thank you for your peer review. The response to your suggestions is below. In addition, the manuscript has been revised. Thank you again for your re-review.
Specific comments and suggestions
Point 1: I am not convinced that the presented data are sufficient to justify a full-length rather than a short report ms.
Response 1: As mentioned below, we are convinced that this data is very important and worth the full paper. However, the final decision is left to the editor.
Point 2: It would be interesting to see the rates of men refused to vaccinate against COVID-19
Not sure why authors focused only in Japanese women aged between 15 to 49 and not older ones, that consist a more vulnerable to COVID-19 group
The comment above also stands for men
Response 2: As shown in the Introduction, females in the age groups of 20–49 years have low vaccine acceptance rates. Vaccination is important for controlling the COVID-19 pandemic. We believe that addressing this specific group, which has a high rate of vaccination hesitation, is an important key to promoting vaccination. I also mentioned in the Materials and Methods section that these are the reasons for focusing on Japanese women aged 15–49. As you’ve stated, we agree that older people are more vulnerable to COVID-19 infection, but we know that they are less hesitant to receive COVID-19 vaccination than are younger people. Vaccination rates for the Japanese elderly are satisfactory.
In Japan, it has been reported that young people and females have large vaccine adverse reactions. Also, as in the case of the HPV vaccine mentioned in the Major Discussion section, many young women will have future pregnancy events. V. Male said at the BMJ: “Vaccine hesitancy among young women is largely driven by false claims that covid-19 vaccines could harm their chances of future pregnancy.”[X] B. Speed referenced by V. Male said: “Young women are the unlikely new face of vaccine resistance/They are now the group most likely to refuse the jab, with many citing misinformation about the vaccine and fertility[Y].” We targeted young Japanese females under the hypothesis that targeted efforts might decrease hesitation regarding COVID-19 vaccination and lead to the control of COVID-19.
I added the following two references on pg. 7 line 207.
[X] V. Male. Menstrual changes after covid-19 vaccination. BMJ 2021; 374
doi: https://doi.org/10.1136/bmj.n2211 (Published 16 September 2021) Cite this as: BMJ 2021;374:n2211
[Y] B. Speed. Young women are the unlikely new face of covid-19 vaccine resistance. i News 2021 Jan 6. https://inews.co.uk/news/health/coronavirus-latest-experts-debunk-vaccine-fertility-myths-women-819783
We are convinced that this data is very important and worth the full paper, again.
Finally, I would like to state that I have submitted my manuscript to English language editing again and improved my English a little.
Sincerely,
Shihoko Kajiwara
School of Nursing, Gifu University of Health Science
2-92 Higashiuzura, Gifu 500-8281, Japan
Phone number: +81-58-274-5001
Email: s-kajiwara@gifuhoken.ac.jp

Reviewer 2 Report
The paper, after minor revision according to the attached report, deserve to be published. Best regards.

Author Response
February 4, 2022
Dear Mr./Ms. peer reviewer 2,
Thank you for your peer review. The response to your suggestions is summarized below. In addition, the manuscript has been revised. Thank you again for your re-review.
Point 1: pg.1 line 22, for a better diffusion of the paper I suggest to write ” intention; healthcare, promoting, hesitancy; Japan.”;
Response 1: Thank you for pointing out the appropriate keywords to diffuse.
I added “healthcare” and “promoting” to the keywords list on pg. 1 line 22.
Point 2: pg.1 line 36, if the authors agree, I suggest to write ”to economic recovery [3–15, X, Y] and reference therein.”, where:
[X] Hara M., Ishibashi M., Nakane A., Nakano T., Hirota Y., Differences in COVID-19 vaccine acceptance, hesitancy, and confidence between healthcare workers and the general population in Japan, Vaccines, 9 (12), art.n. 1389, (2021);
[Y] Ragusa R., Bertino G., Bruno A., Frazzetto E., Cicciu F., Giorgianni G., Lupo L., Evaluation of health status in patients with hepatitis c treated with and without interferon, Health and quality of life ourcomes, 16, art.n. 17, (2018);
Response 2: Thank you for suggesting these interesting papers. We agree that these are important.
I have added the above two references on pg. 1 line 36.
Point 3: pg.4 line 16, ”allergic to vaccines and women that have other physical reasons” are treated in the same section but could be interesting in a future research to share these two answers because in the other could be also the bad willingness to do the vaccination.
Response 3: Reasons categorized as “I am allergic to vaccines/I have other physical reasons” include “I am allergic(10)/ I am pregnant(5)/ I am sick(3)/ I can't go to the mass vaccination site because I have a mental illness/ I have a chronic disease/ and so on.” I will share some of these reasons on pg.4 lines 135–137. Off course the full data are available on request as stated in the Data Availability Statement; however, it is in Japanese.
Point 4: pg.7 line 162, I suggest to delete Discussions and write Major discussion because this section is followed by a Conclusions section where, in effect, the final balance is clearly written and the final considerations are expressed by the authors.
Response 4: Thank you for your suggestion. I have deleted “Discussions” and wrote “Major discussion” on pg.7 line 174.
Point 5: pg.8 line 233, please, spend some words to explain the limitations. In my opinion all is too much short described, more details are appreciated.
Response 5: We apologize for the lack of explanation.
I added an explanation with an example on page 8 lines 250–256.
Finally, I would like to state that I have submitted my manuscript to English language editing again and improved my English a little.
Sincerely,
Shihoko Kajiwara
School of Nursing, Gifu University of Health Science
2-92 Higashiuzura, Gifu 500-8281, Japan
Phone number: +81-58-274-5001
Email: s-kajiwara@gifuhoken.ac.jp

Round 2
Reviewer 1 Report
i have no further comments
Author Response
Dear Mr./Ms. peer reviewer 1,
I apologize for misunderstanding your comments.
Your comments are completely right.
We will expand the analysis in a wider population in a future study.
According to your suggestions, this time we would like to submit the manuscript as a short Communication.
Thank you for your advice.
Sincerely,
Shihoko Kajiwara
Email: s-kajiwara@gifuhoken.ac.jp